# Kinetic field theory: Non-linear cosmic power spectra in the mean-field approximation

Matthias Bartelmann[1*], Johannes Dombrowski[2], Sara Konrad[1], Elena Kozlikin[1], Robert Lilow[3], Carsten Littek[1], Christophe Pixius[1], Felix Fabis[4]

**1** Institute for Theoretical Physics, Heidelberg University, Germany
**2** School of Physics and Astronomy, University of Nottingham, UK
**3** Department of Physics, Technion, Haifa, Israel
**4** Institute for Theoretical Astrophysics, ZAH, Heidelberg University, Germany
* bartelmann@uni-heidelberg.de

November 11, 2020

## Abstract

**We use the recently developed Kinetic Field Theory (KFT) for cosmic structure formation to show how non-linear power spectra for cosmic density fluctuations can be calculated in a mean-field approximation to the particle interactions. Our main result is a simple, closed and analytic, approximate expression for this power spectrum. This expression has two parameters characterising non-linear structure growth which can be calibrated within KFT itself. Using this self-calibration, the non-linear power spectrum agrees with results obtained from numerical simulations to within typically $\lesssim 10\,\%$ up to wave numbers $k \lesssim 10\,h\,\mathrm{Mpc}^{-1}$ at redshift $z = 0$. Adjusting the two parameters to optimise agreement with numerical simulations, the relative difference to numerical results shrinks to typically $\lesssim 5\,\%$. As part of the derivation of our mean-field approximation, we show that the effective interaction potential between dark-matter particles relative to Zel'dovich trajectories is sourced by non-linear cosmic density fluctuations only, and is approximately of Yukawa rather than Newtonian shape.**

# 1 Introduction

Kinetic Field Theory (KFT) describes ensembles of classical particles in and out of equilibrium [1–3]. It is based upon the Martin-Siggia-Rose approach to classical statistical systems [4] and has been adapted to cosmological initial conditions and to the expanding cosmological background in previous papers [5, 6]. Its central mathematical object is a generating functional encapsulating the statistical properties of the initial state, the Green's function or propagator of the equations of motion, and the particle-particle interactions. These interactions are described by an exponential operator acting on the free generating functional. In the conventional approach to statistical field theories, this operator is expanded into a Taylor series, leading to a systematic approach to perturbation theory in terms of Feynman diagrams.

In this paper, we show that the interaction operator can instead be approximated as an averaged interaction term using a mean-field approach. This can be done in such a way that its action on the generating functional can be separated from the integration over the initial phase-space distribution. This results in a numerical, time and scale-dependent factor multiplying the free generating functional. Averaging over a pair of density factors then leads to an approximate, but closed and analytic expression for the non-linear power spectrum of cosmic density fluctuations.

The mean-field approximation introduces two parameters, the non-linear scale and an effective viscosity reducing the velocity variance after shell-crossing. Both of them can be calibrated from within KFT itself. With these parameters self-calibrated in this way, our mean-field approximation to the non-linear power spectrum agrees with results from numerical simulations with a relative deviation of typically $\lesssim 10\%$ up to $k \approx 10\,h\,\mathrm{Mpc}^{-1}$ at redshift $z = 0$. Alternatively, these parameters can be optimised to further improve the agreement between the non-linear power spectra from our mean-field approximation and from numerical simulations. Doing so, the relative deviation to numerical results for $\Lambda$CDM can be lowered to $\lesssim 5\,\%$ in the same range of wave numbers.

In Sect. 2, we discuss the trajectories of Hamiltonian particles in the expanding cosmic space-time. Using results derived in detail in Appendix A, we show that the conventional Zel'dovich approximation

for the inertial motion of particles implies that the remaining particle-interaction potential is sourced only by the non-linearly evolved density contrast, which leads to an approximately Yukawa-shaped cut-off of the Newtonian gravitational potential. In Sect. 3, we briefly review the calculation of power spectra from kinetic field theory. In Sect. 4, we develop our mean-field approach to the particle-particle interaction term, and we summarise and discuss our results in Sect. 5.

We use the convention

$$\mathcal{F}[f] =: \tilde{f}(\vec{k}) = \int_q f(\vec{q}) \, \mathrm{e}^{-\mathrm{i}\vec{k}\cdot\vec{q}} \,, \quad \mathcal{F}^{-1}\left[\tilde{f}\right] = f(\vec{q}) = \int_k \tilde{f}(\vec{k}) \, \mathrm{e}^{\mathrm{i}\vec{k}\cdot\vec{q}} \tag{1}$$

for the Fourier transform $\mathcal{F}$ and its inverse $\mathcal{F}^{-1}$, with the short-hand notations

$$\int_q := \int \mathrm{d}^3 q \,, \quad \int_k := \int \frac{\mathrm{d}^3 k}{(2\pi)^3} \,. \tag{2}$$

Where needed, we adopt a (spatially flat) $\Lambda$CDM cosmological model with matter-density parameter $\Omega_{\mathrm{m0}} = 0.3$.

## 2 Particle dynamics

### 2.1 Particle trajectories

On the expanding spatial background of a Friedmann-Lemaître model universe with scale factor $a$, we introduce comoving coordinates $\vec{q}$ and use the linear growth factor $D_+$ as the time coordinate $t$. We set initial conditions at some time $t_{\mathrm{i}}$ in the distant past when matter just began dominating the dynamics of the cosmic expansion.

For later convenience, we set both the scale factor $a$ and the growth factor $D_+$ to unity at the initial time such that $t = D_+ - 1$ and $t_{\mathrm{i}} = 0$ initially. In these coordinates, the Hamiltonian of point particles is

$$\mathcal{H} = \frac{\vec{p}^2}{2m} + m\varphi \tag{3}$$

with an effective, dimension-less, time-dependent particle mass $m$ given by

$$m = a^3 D'_+(a)E(a) \,, \quad E(a) := \frac{H(a)}{H_{\mathrm{i}}} \tag{4}$$

in terms of the derivative $D'_+(a) = \mathrm{d}D_+(a)/\mathrm{d}a$ of the linear growth factor and the expansion function $E(a)$, defined as the Hubble function $H(a)$ divided by the Hubble constant $H_{\mathrm{i}}$ at the initial time. The particle mass $m$ used to be called $g$ in our earlier papers on KFT, but we change the notation here to acquire a more intuitive meaning. Like $a$ and $D_+$, the expansion function $E$ is supposed to be normalised to unity at the initial time such that $m(t_{\mathrm{i}}) = 1$. In an Einstein-de Sitter universe, $D_+ = a$ and $E = a^{-3/2}$, thus $m = a^{3/2} = (1 + t)^{3/2}$. The potential $\varphi$ in (3) satisfies the Poisson equation

$$\vec{\nabla}^2 \varphi = A_\varphi \delta \quad \text{with} \quad A_\varphi := \frac{3}{2} \Omega_{\mathrm{m}}^{(\mathrm{i})} \frac{a}{m^2} \tag{5}$$

sourced by the density contrast $\delta$ relative to the background density [7]. The Laplacian in (5) acts with respect to the comoving coordinates $\vec{q}$.

Combining the phase-space coordinates of an individual particle $j$ into a vector $\vec{x}_j := (\vec{q}_j, \dot{\vec{q}}_j)^{\top}$, the solution of the Hamiltonian equation of motion beginning at $\vec{x}_j^{(i)} = (\vec{q}_j^{(i)}, \vec{p}_j^{(i)})$ at $t_i$ can be written in the form

$$\vec{x}_j(t) = G(t, 0)\vec{x}_j^{(i)} + \int_0^t \mathrm{d}t' \, G(t, t') \begin{pmatrix} 0 \\ \vec{f}_j(t') \end{pmatrix} \tag{6}$$

with the matrix-valued propagator $G(t, t')$ and the effective force $\vec{f}_j(t')$ on particle $j$, introduced and derived in Appendix A.

Instead of individual particles, we consider canonical ensembles of $N \gg 1$ classical particles $j$ at positions $\vec{q}_j$ with velocities $\dot{\vec{q}}_j$ and introduce the tensor $\boldsymbol{x} = \vec{x}_j \otimes \vec{e}_j$ to bundle the phase-space coordinates of the entire ensemble, where $\vec{e}_j$ is the $j$-th Cartesian unit vector in $N$ dimensions. This allows us to write the entire bundle of all particle trajectories in the compact form

$$\boldsymbol{x}(t) = \boldsymbol{G}(t, 0)\boldsymbol{x}^{(i)} + \int_0^t \mathrm{d}t' \, \boldsymbol{G}(t, t') \begin{pmatrix} 0 \\ \boldsymbol{f}(t') \end{pmatrix} =: \boldsymbol{x}_0(t) + \boldsymbol{y}(t) \tag{7}$$

with $\boldsymbol{G}(t, t') := G(t, t') \otimes \mathbb{1}_N$ and $\boldsymbol{f}(t') := \vec{f}_j(t') \otimes \vec{e}_j$. The inertial trajectories are $\boldsymbol{x}_0 = \boldsymbol{G}(t, 0)\boldsymbol{x}^{(i)}$, and $\boldsymbol{y}$ are the deviations therefrom caused by the effective force $\boldsymbol{f}$.

## 2.2 Effective gravitational potential

The potential of the effective force acting relative to Zel'dovich trajectories is given by Eq. (73) of Appendix A,

$$\phi = \varphi + A_\varphi D_+ \psi \,, \tag{8}$$

where $\psi$ is the potential of the curl-free initial velocity field. With the Poisson equations (5) for $\varphi$ and $\vec{\nabla}^2 \psi = -\delta^{(i)}$ for the initial velocity potential $\psi$, the Poisson equation for $\phi$ is

$$\vec{\nabla}^2 \phi = A_\varphi \left( \delta - D_+ \delta^{(i)} \right) = A_\varphi \left( \delta - \delta^{(\mathrm{lin})} \right) \,, \tag{9}$$

where $\delta^{(\mathrm{lin})} = D_+ \delta^{(i)}$ is the linearly growing density contrast. The potential $\phi$ describing the gravitational interaction relative to the inertial particle trajectories $\boldsymbol{x}_0(t)$ is thus sourced exclusively by the non-linearly evolved contribution to the density fluctuations. The effective gravitational force mediated by $\phi$ is therefore confined to small scales.

This has important consequences for kinetic theory. If we describe inertial particle orbits with the Zel'dovich propagator, their mutual interaction must be modified in such a way that only the non-linear density contrast contributes to the force between them. This reflects the fact that the Zel'dovich propagator already takes the large-scale part of the gravitational interaction between the particles into account. Since the Zel'dovich trajectories reflect effective inertial motion with respect to the time coordinate $t = D_+ - 1$, and since the gravitational force caused by the linear density contrast relative to these trajectories needs to vanish, only the *deviation* of the density contrast from its linear value can be the source of the effective gravitational interaction. On large scales, where the density contrast keeps growing linearly for all cosmologically relevant times, and where the Zel'dovich approximation describes the particle motion accurately, the effective force must vanish. On small scales, where non-linear structures build up, the effective gravitational interaction must set in as non-linear density contrasts develop.

## 2.3 Shape of the effective potential

The effective gravitational potential between particles following Zel'dovich trajectories must thus deviate from the Newtonian form in such a scale-dependent way that the force tends to zero for $k \ll k_0$ and approaches the Newtonian form for $k \gg k_0$, with the wave number $k_0$ set by the time-dependent boundary between linear and non-linear scales.

For quantifying how the potential needs to be modified, we search for a particle-particle interaction potential $v$ such that the collective potential $\phi = n\delta * v$ of the particle ensemble with the mean number density $n$, i.e. the convolution of the particle number-density fluctuation $n\delta$ with the potential $v$, satisfies the Poisson equation (9). Using the Fourier convolution theorem, the Fourier transform of this Poisson equation reads

$$\tilde{\delta}\,\tilde{v} = -\frac{A_\varphi}{nk^2}\left(\tilde{\delta} - \tilde{\delta}^{(\mathrm{lin})}\right) . \tag{10}$$

Multiplying this equation once with $\tilde{\delta}$, once with $\tilde{\delta}^{(\mathrm{lin})}$, and taking the ensemble average gives

$$P_\delta\tilde{v} \approx -\frac{A_\varphi}{nk^2}\left(P_\delta - \left\langle\tilde{\delta}\tilde{\delta}^{(\mathrm{lin})}\right\rangle\right) \quad \text{and}$$

$$\left\langle\tilde{\delta}\tilde{\delta}^{(\mathrm{lin})}\right\rangle\tilde{v} \approx -\frac{A_\varphi}{nk^2}\left(\left\langle\tilde{\delta}\tilde{\delta}^{(\mathrm{lin})}\right\rangle - P_\delta^{(\mathrm{lin})}\right) , \tag{11}$$

introducing the power spectrum $P_\delta$ of the density contrast. Going from (10) to (11), we have implicitly assumed that the form of $\tilde{v}$ is independent of the density contrast, which should be a good approximation, but does not generally need to be the case. Eliminating $\left\langle\tilde{\delta}\tilde{\delta}^{(\mathrm{lin})}\right\rangle$ between these equations leads to a quadratic equation for $\tilde{v}$ whose only meaningful solution is

$$\tilde{v} = -\frac{A_\varphi}{nk^2}\,f_v(k) \quad \text{with} \quad f_v(k) = 1 - \left(\frac{P_\delta^{(\mathrm{lin})}}{P_\delta}\right)^{1/2} . \tag{12}$$

The function $f_v(k)$ turns to zero for wave numbers $k$ small enough to fall into the linear regime, and to unity for $k$ large enough to be deeply in the non-linear regime. Power spectra obtained from numerical simulations [8–10] suggest that the function

$$f_v(k) = \frac{k^2}{k_0^2 + k^2} \tag{13}$$

represents the transition from large to small scales reasonably well, with $k_0$ quantifying the wave number above which non-linear evolution begins to dominate (see Fig. 1). With (12), this results in the effective potential

$$\tilde{v} = -\frac{A_\varphi}{n\left(k_0^2 + k^2\right)} , \tag{14}$$

which is of Yukawa rather than Newtonian form.

Note that the expression (12) for the particle-particle interaction potential $v$ is statistical, i.e. it depends on the spatial correlations within the particle ensemble.

The essential result of this discussion is thus that the effective interaction between particles following Zel'dovich trajectories is mediated by an approximately Yukawa-like rather than a Newtonian potential. It is important for our purposes to note that the scale $k_0$ can be determined from KFT itself in a way to be described in Sect. 4.3. The right panel in Fig. 1 shows the time-dependent Yukawa scale $k_0$ determined in this way.

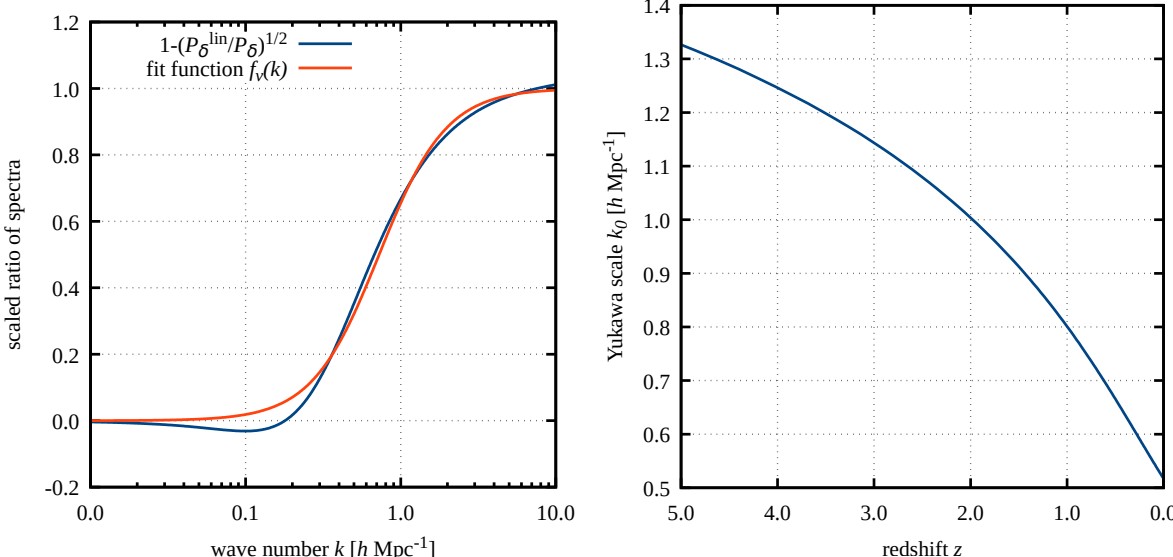

Figure 1: *Left*: The ratio of power spectra $1 - (P_\delta^{(\mathrm{lin})}/P_\delta)^{1/2}$, scaled to unity at $k \gg 1$, compared to the fitting function $f_v(k)$ from Eq. (13). The linear power spectrum was taken from [11], the non-linear from [9]. *Right*: The Yukawa scale $k_0$ as a function of the redshift $z$.

## 3 Power spectra from KFT

We briefly review in this section the KFT approach to cosmic structure formation. For further detail, we refer the reader to [5], [12], and the review [6].

### 3.1 Generating functional

The central mathematical object of KFT is its generating functional $Z$. Like a partition sum in thermodynamics, it is a phase-space integral over the probability $P(x)$ for the phase-space positions $x$ to be occupied. Splitting $P(x)$ into a probability $P(x^{(i)})$ for an initial state times a transition probability $P(x|x^{(i)})$ from the initial to the final state, further introducing a generator field

$$J := \begin{pmatrix} \vec{J}_{q_i}(t) \\ \vec{J}_{p_i}(t) \end{pmatrix} \otimes \vec{e}_i \tag{15}$$

to allow extracting moments of particle positions and momenta via functional derivatives with respect to $J$ later, and introducing the particle trajectories $x(t)$ from (7), leads to the generating functional

$$Z[J] = \int d\Gamma \, e^{iJ \cdot x} \tag{16}$$

derived in [12], with the dot denoting the time-integrated scalar product

$$A \cdot B = \int dt \, \left\langle \vec{A}_i, \vec{B}_i \right\rangle \, . \tag{17}$$

According to (7), the phase in (16) can be split into a free and an interacting part,

$$Z[J] = \int d\Gamma \, e^{iJ \cdot (x_0 + y)} \, . \tag{18}$$

Since the contribution $S_{\mathrm{I}}[J] := \mathrm{i}J \cdot y$ to the phase depends on all relative positions of the correlated particle ensemble, it seems impossible to evaluate the generating functional $Z[J]$ analytically. A systematic approach to perturbation theory may begin with converting the interaction term into an operator acting on the free generating functional $Z_0$ defined in (20) below, followed by Taylor-expanding this operator. We have previously shown that even the first order of this perturbative approach leads to non-linear density-fluctuation power spectra close to results from numerical simulations [5], and we will further analyse KFT perturbation theory in future papers. Here, we follow a different path.

The essential purpose of this paper is to find a suitable average for the interacting part,

$$S_{\mathrm{I}}[J] \rightarrow \langle S_{\mathrm{I}}[J] \rangle \ , \tag{19}$$

which would allow us to write the generating functional (18) as

$$Z[J] \approx \mathrm{e}^{\langle S_{\mathrm{I}}[J] \rangle} \int \mathrm{d}\Gamma\, \mathrm{e}^{\mathrm{i}J \cdot x_0} =: \mathrm{e}^{\langle S_{\mathrm{I}}[J] \rangle} Z_0[J] \ . \tag{20}$$

Before we proceed to construct and analyse such an average, we briefly review how power spectra are derived in KFT from the generating functional $Z[J]$.

## 3.2 Density cumulants

The particle number density is a sum of delta distributions centered on the particle positions at time $t$ or, in a Fourier representation,

$$\tilde{\rho}\left(\vec{k}, t\right) = \sum_{i=1}^{N} \mathrm{e}^{-\mathrm{i}\vec{k} \cdot \vec{q}_i(t)} =: \sum_{i=1}^{N} \tilde{\rho}_i\left(\vec{k}, t\right) \ . \tag{21}$$

Replacing the particle position $\vec{q}_i$ in each one-particle density contribution $\tilde{\rho}_i(\vec{k}, t)$ by a functional derivative with respect to $\vec{J}_{q_i}(t)$, we obtain the one- and $N$-particle density operators

$$\hat{\rho}_i\left(\vec{k}, t\right) = \exp\left(-\mathrm{i}\vec{k} \cdot \frac{\delta}{\mathrm{i}\delta\vec{J}_{q_i}(t)}\right) , \quad \hat{\rho}\left(\vec{k}, t\right) = \sum_{i=1}^{N} \hat{\rho}_i\left(\vec{k}, t\right) \ . \tag{22}$$

Since the density operators are exponentials of derivatives with respect to components of $J$, they generate translations of the generating functional, $Z[J] \rightarrow Z[J + L]$.

Density cumulants of order $r$ are obtained by applying $r$ density operators to the generating functional. For synchronous power spectra, i.e. cumulants of order $r = 2$ with $t_1 = t_2 =: t$, we have

$$G_{\rho\rho}(1, 2) = \sum_{i \neq j=1}^{N} \hat{\rho}_i(1)\hat{\rho}_j(2)\, Z[J] = \sum_{i \neq j=1}^{N} Z[J + L] \ , \tag{23}$$

with the corresponding shift

$$L = -\delta_{\mathrm{D}}(t' - t)\begin{pmatrix} 1 \\ 0 \end{pmatrix}\left(\vec{k}_1 \otimes \vec{e}_i + \vec{k}_2 \otimes \vec{e}_j\right) \ . \tag{24}$$

The short-hand notation $(n)$ for the arguments in (23) indicates the Fourier-space position $(\vec{k}_n, t_n)$ at time $t_n$. The generator field $J$ can be set to zero once all density operators have acted on the generating functional. Since the particles of the ensemble are indistinguishable, each term under the sum in (23)

gives the same result as for any particle pair arbitrarily labelled as $i, j = 1, 2$, and the cumulant becomes

$$G_{\rho\rho}(1, 2) = N(N - 1) Z[\boldsymbol{L}] \approx N(N - 1) \, \mathrm{e}^{\langle S_1[\boldsymbol{L}]\rangle} Z_0[\boldsymbol{L}] \, . \tag{25}$$

With initial conditions appropriate for the early universe, the free generating functional $Z_0[\boldsymbol{J}]$ after applying two density operators $\hat{\rho}_1(1)$ and $\hat{\rho}_2(2)$ can be written as

$$Z_0[\boldsymbol{L}] = (2\pi)^3 \delta_{\mathrm{D}}\left(\vec{k}_1 + \vec{k}_2\right) V^{-2} \mathrm{e}^{-Q_{\mathrm{D}}} \, \mathcal{P}(k_1) =: (2\pi)^3 \delta_{\mathrm{D}}\left(\vec{k}_1 + \vec{k}_2\right) V^{-2} \bar{\mathcal{P}}(k_1) \tag{26}$$

with the damping term

$$Q_{\mathrm{D}} := \frac{k^2}{3} (\sigma_1 t)^2 \tag{27}$$

and the freely evolved, non-linear power spectrum

$$\mathcal{P}(k_1) := \int_q \left[\mathrm{e}^{-t^2 k_1^2 a_\parallel(q,\mu)} - 1\right] \mathrm{e}^{\mathrm{i}\vec{k}_1 \cdot \vec{q}} \, ; \tag{28}$$

see [12] for the derivation. The function $a_\parallel$ appearing here is the auto-correlation function of momentum components parallel to the wave vector $\vec{k}_1$,

$$a_\parallel = \mu^2 \xi_\psi''(q) + \left(1 - \mu^2\right) \frac{\xi_\psi'(q)}{q} \, , \tag{29}$$

where $\mu$ is the direction cosine of $\vec{q}$ relative to $\vec{k}_1$. The function

$$\xi_\psi(q) = \frac{1}{2\pi^2} \int_0^\infty \frac{\mathrm{d}k}{k^2} P_\delta^{(\mathrm{i})}(k) j_0(kq) \tag{30}$$

containing the spherical Bessel function $j_0$ is the auto-correlation function of the initial velocity potential $\psi$, determined by the initial density-fluctuation power spectrum $P_\delta^{(\mathrm{i})}(k)$. The initial velocity field is supposed to be the gradient of a velocity potential because any initial curl would decay quickly due to cosmic expansion and angular-momentum conservation. We further define the moments

$$\sigma_n^2 := \int_k k^{2(n-2)} P_\delta^{(\mathrm{i})}(k) = \frac{1}{2\pi^2} \int_0^\infty \mathrm{d}k \, k^{2n-2} P_\delta^{(\mathrm{i})}(k) \tag{31}$$

of the density-fluctuation power spectrum and note that

$$\lim_{q \to 0} a_\parallel = -\frac{\sigma_1^2}{3} \, . \tag{32}$$

For small arguments of the first exponential in (28), $\mathcal{P}$ turns into the linearly evolved power spectrum,

$$\mathcal{P}(k) \to (1 + t)^2 P_\delta^{(\mathrm{i})}(k) = P_\delta^{(\mathrm{lin})}(k) \, , \tag{33}$$

as shown in [12].

## 3.3 Damping and interaction

The damping term $Q_D$ in (26) requires a careful discussion. Its definition (27) in conjunction with the velocity dispersion $\sigma_1^2$ from (31) shows that it arises because particles stream freely with an average velocity $\sigma_1$ in the Zel'dovich approximation. We should emphasise that this velocity dispersion is not of thermal origin, but arises from drawing initial particle velocities from a velocity potential which is a homogeneous and isotropic Gaussian random field. Where this velocity field converges, structures form, but these structures are smoothed in a system of free particles once caustics have been formed and converging particle streams have crossed.

In (26), it appears as if the damping term would exponentially reduce the power. However, this is not the case. Rather, numerical integration and asymptotic analysis alike show that the freely-evolved power spectrum $\bar{\mathcal{P}}$ *combined* with the damping term follows the linearly evolved power spectrum $P_\delta^{(\text{lin})}$ on large scales, drops below it on non-linear scales, but turns towards an asymptotic behaviour $\propto k^{-3}$ as $k$ increases further [12].

Combining (25) and (26) to

$$G_{\rho\rho}(1,2) \approx n^2 \mathrm{e}^{\langle S_I[L]\rangle - Q_D} \mathcal{P}(k_1) \tag{34}$$

shows that the gravitational interaction between the particles counteracts this characteristic reduction of the power. In fact, a major part of the interaction term is required for keeping structures in place once they have formed. Any surplus, i.e. any positive difference between $\langle S_I\rangle$ and $Q_D$, leads to non-linear structure growth.

# 4 Mean-field approach to non-linear power spectra

Based on these arguments, we now pursue the following approach. We wish to represent the particle interactions by a suitably averaged interaction term $\langle S_I[L]\rangle$. For simplicity, we further wish to approximate the damped, freely evolving power spectrum $\bar{\mathcal{P}}$ by the linearly evolving power spectrum $P_\delta^{(\text{lin})}$. As just discussed, this implies that we are ignoring the reduction of power by the velocity variance after shell crossing. Since the interaction term counter-acts the damping, we then also need to reduce the interaction term on non-linear scales. We will do so by using Burgers' approximation. The result will be the simple expression

$$P_\delta^{(\text{nl})}(k) \approx \mathrm{e}^{\langle S_I\rangle(k)} P_\delta^{(\text{lin})}(k) \tag{35}$$

for the non-linear power spectrum. Our main result will be a specific and simple equation for $\langle S_I\rangle(k)$ reproducing numerically derived, non-linear power spectra remarkably well.

## 4.1 Averaged particle-particle force

With $\boldsymbol{y}$ from (7) and $\boldsymbol{L}$ from (24), we have

$$S_I[\boldsymbol{L}] = \mathrm{i}\boldsymbol{L}\cdot\boldsymbol{y} = -\mathrm{i}\int_0^t \mathrm{d}t'\,(t-t')\left[\vec{k}_1\cdot\vec{f}_1(t') + \vec{k}_2\cdot\vec{f}_2(t')\right] \tag{36}$$

where $\vec{f}_i(t)$ is the effective force on an arbitrary particle $i$. Note again that the time $t$ here is not the cosmological time, but defined by the linear growth factor $D_+$. In terms of an effective particle-particle

interaction force $\vec{f}_{\mathrm{p}}$ and the particle number density $\rho$, we can write the force $\vec{f}_i$ on particle $i$ as

$$\vec{f}_i = \int_{q_1} \int_{q_2} \rho_i(\vec{q}_1) \vec{f}_{\mathrm{p}}(\vec{q}_1 - \vec{q}_2) \rho(\vec{q}_2) \ . \tag{37}$$

We now average this force term over particle ensembles drawn from a statistically homogeneous, correlated random density field. Since we wish to retain the dependence of the resulting, averaged, effective force term $\langle \vec{f}_i \rangle$ on the wave vector $\vec{k}$, we project out the contribution by the mode $\vec{k}$ of the density field, writing

$$\langle \vec{f}_i \rangle(\vec{k}) = \int_{q_1} \int_{q_2} \vec{f}_{\mathrm{p}}(\vec{q}_1 - \vec{q}_2) \langle \rho_i(\vec{q}_1) \rho(\vec{q}_2) \rangle \, e^{-i\vec{k}\cdot(\vec{q}_1 - \vec{q}_2)} \ . \tag{38}$$

The average over the product of densities introduces the correlation function $\xi(|\vec{q}_1 - \vec{q}_2|)$ of the density field,

$$\langle \rho_i(\vec{q}_1) \rho(\vec{q}_2) \rangle = \frac{1}{N} \langle \rho(\vec{q}_1) \rho(\vec{q}_2) \rangle = \frac{n^2}{N} \left[ 1 + \xi\left( |\vec{q}_1 - \vec{q}_2| \right) \right] \ . \tag{39}$$

We keep only the connected part of the correlation expressed by $\xi$ alone in (39) because the disconnected part cannot contribute in a homogeneous random field. Owing to homogeneity, the integrand in (38) depends only on the difference $\vec{q}_1 - \vec{q}_2 =: \vec{q}$ of position vectors. We can thus integrate over $\vec{q}_1$, resulting in a factor $V$, and obtain

$$\langle \vec{f}_i \rangle(\vec{k}) = n \int_q \vec{f}_{\mathrm{p}}(\vec{q}) \xi(q) e^{-i\vec{k}\cdot\vec{q}} \ . \tag{40}$$

Since the remaining expression is the Fourier transform of a product, the scale-dependent, mean force term is the convolution of the particle-particle force $\tilde{f}_{\mathrm{p}}$ in Fourier space and the power spectrum $P_\delta$ of the particle distribution as the Fourier transform of the correlation function,

$$\langle \vec{f}_i \rangle(\vec{k}) = n \left( \tilde{f}_{\mathrm{p}} * P_\delta \right)(\vec{k}) \ . \tag{41}$$

Combining (41) with (36) and using Newton's third axiom in the form $\langle \vec{f}_i \rangle(-\vec{k}) = -\langle \vec{f}_i \rangle(\vec{k})$, we thus find the expression

$$\langle S_I \rangle(\vec{k}) = -2in\vec{k} \cdot \int_0^t \mathrm{d}t' \, (t - t') \left( \tilde{f}_{\mathrm{p}} * P_\delta \right)(\vec{k}) \tag{42}$$

for the averaged, scale-dependent, interaction term.

## 4.2 Damping within Burgers' approximation

We now need to specify the power spectrum $P_\delta$ to be inserted into (41) for evaluating the average force term $\langle \vec{f}_i \rangle$. Following (38), evaluating the mean interaction term with the density correlation function suggests replacing $P_\delta = \bar{\mathcal{P}}$. Applying the same linear approximation leading from (34) to (35), we would then arrive at $P_\delta \approx P_\delta^{(\mathrm{lin})}$. However, since the linear power spectrum in (35) ignores damping and thus overestimates the power on non-linear scales, we need to reduce the average force term on these scales appropriately. The amount of this reduction can be effectively estimated by Burgers' approximation [13–15].

Burgers' approximation changes the inertial motion of particles in the Zel'dovich time coordinate in a way derived from the Navier-Stokes equation of hydrodynamics,

$$\frac{\mathrm{d}\dot{\vec{q}}}{\mathrm{d}t} = 0 \quad \rightarrow \quad \frac{\mathrm{d}\dot{\vec{q}}}{\mathrm{d}t} = \nu \vec{\nabla}^2 \dot{\vec{q}} \ , \tag{43}$$

where $\nu$ is a viscosity parameter with the dimension of a squared length. A natural choice for the length scale $\nu^{1/2}$ is the non-linear radius $r_{\mathrm{nl}}$ defined by

$$\sigma^2_{r_{\mathrm{nl}}} = \int_k P_\delta(k) W^2_R(k)\Big|_{R=r_{\mathrm{nl}}} = 1 \tag{44}$$

evaluated with the linear power spectrum from [11] and a top-hat window function $W_R(k)$ at the present cosmic time. We set $\nu = r^2_{\mathrm{nl}} = \mathrm{const}$ here for simplicity, but note that $\nu$ could also be generalised to become time-dependent.

Burgers' equation can be solved by a Hopf-Cole transformation [16, 17], which results in

$$\dot{\vec{q}} = -2\nu\vec{\nabla}\ln U \;, \tag{45}$$

where $U$ is an exponential velocity potential given by the convolution

$$U = \mathcal{N}_{\sqrt{2\nu t}} * \exp\left(-\frac{\psi}{2\nu}\right) \tag{46}$$

of the scaled, exponentiated initial velocity potential $\psi$ with a normal distribution $\mathcal{N}$ of width $(2\nu t)^{1/2}$. To linear order in $\psi$, (45) implies the velocity dispersion

$$\sigma^2_v(t) = \int_k \frac{P_\delta(k)}{k^2}\exp\left(-2k^2\nu t\right)\;. \tag{47}$$

This expression clearly shows the effect of Burgers' approximation: small-scale modes with $k \gtrsim (2\nu t)^{-1/2}$ are removed from the velocity field, slowing down the particles on such scales and thus reducing the re-expansion of structures after stream crossing [18–21]. We take this reduced amount of particle motion into account by replacing the damping term $Q_D$ from (27) by

$$\bar{Q}_D = k^2\lambda^2 \quad \text{with the damping scale} \quad \lambda(t) = \int_0^t \mathrm{d}t'\,\sigma_v(t')\;. \tag{48}$$

An excellent fit to $\lambda(t)$ is

$$\lambda(t) \approx \frac{t}{\sqrt{1+t/\tau}} \quad \text{with} \quad \tau \approx 24.17\;. \tag{49}$$

The form of this fit expresses the transition from ballistic to diffusive particle motion.

## 4.3  Averaged interaction term

Accordingly, we evaluate the averaged force term (41) as

$$\left\langle\vec{f_i}\right\rangle\!\left(\vec{k}\right) = n\left(\tilde{f}_{\mathrm{p}} * \bar{P}_\delta\right)(k) \quad \text{with} \quad \bar{P}_\delta(k) = \left(1 + \bar{Q}_D\right)^{-1} P^{(\mathrm{lin})}_\delta(k)\;. \tag{50}$$

Inserting the averaged force term (50) into (42) results in the averaged interaction term

$$\left\langle S_{\mathrm{I}}\right\rangle(k) = -2\mathrm{i}n\vec{k}\cdot\int_0^t \mathrm{d}t'\,(t-t')\left(\tilde{f}_{\mathrm{p}} * \bar{P}_\delta\right)(k)\;. \tag{51}$$

We finally need to evaluate the convolution of the particle-particle force term $\tilde{f}_{\mathrm{p}}$ with the damped power spectrum $\bar{P}$. This is done in Appendix B and results in the average interaction term

$$\left\langle S_{\mathrm{I}}\right\rangle(k) = 2\int_0^t \mathrm{d}t'\,(t-t')\frac{\dot{m}}{m}\left[D_+\sigma^2_J - \frac{1}{m}\int_0^{t'} \mathrm{d}\bar{t}\,\dot{m}D_+\sigma^2_J\right]\;, \tag{52}$$

where $\sigma_J^2$ is the moment (83) of the damped power spectrum. This interaction term $\langle S_I \rangle$ in the mean-field approximation is shown in the right panel of Fig. 4 as a function of wave number $k$ for redshift $z = 0$. As it has to be, the averaged interaction term is dimension-less.

The scale $k_0$ of the Fourier-transformed Yukawa-like potential (14) still needs to be set. KFT itself now suggests the following procedure. According to (35), the ratio between the linearly and non-linearly evolved power spectra is estimated by the exponential of the averaged interaction term. Combining (35) with (12), the factor $f_v$ from (12) is approximated by

$$f_v(k) \approx 1 - e^{-\langle S_I \rangle / 2} \ . \tag{53}$$

We can thus determine a first estimate for $k_0$ by calculating $\langle S_I \rangle$ with $k_0 = 0$ and fitting $f_v(k)$ from (53) with the functional form (13). With the resulting value of $k_0$, an updated estimate for $\langle S_I \rangle$ can be calculated, and so forth. This iteration quickly converges; it turns out that even one step suffices. The scale $k_0$ determined from KFT in this way is shown in the right panel of Fig. 1 as a function of redshift $z$. For simplicity, we adopt the constant value of $k_0$ at $z = 0$ and ignore its time dependence.

## 4.4 Non-linear density-fluctuation power spectrum

The mean-field averaged interaction term (52), inserted into (35), is our approximate expression for the non-linear density-fluctuation power spectrum. Evaluating the averaged interaction term $\langle S_I \rangle(k)$ from (52) numerically, determining the viscosity parameter $\nu$ and the Yukawa scale $k_0$ from KFT itself as described, and further assuming a spatially-flat $\Lambda$CDM model universe with matter-density parameter $\Omega_{m0} = 0.3$, leads to the result shown in Fig. 2.

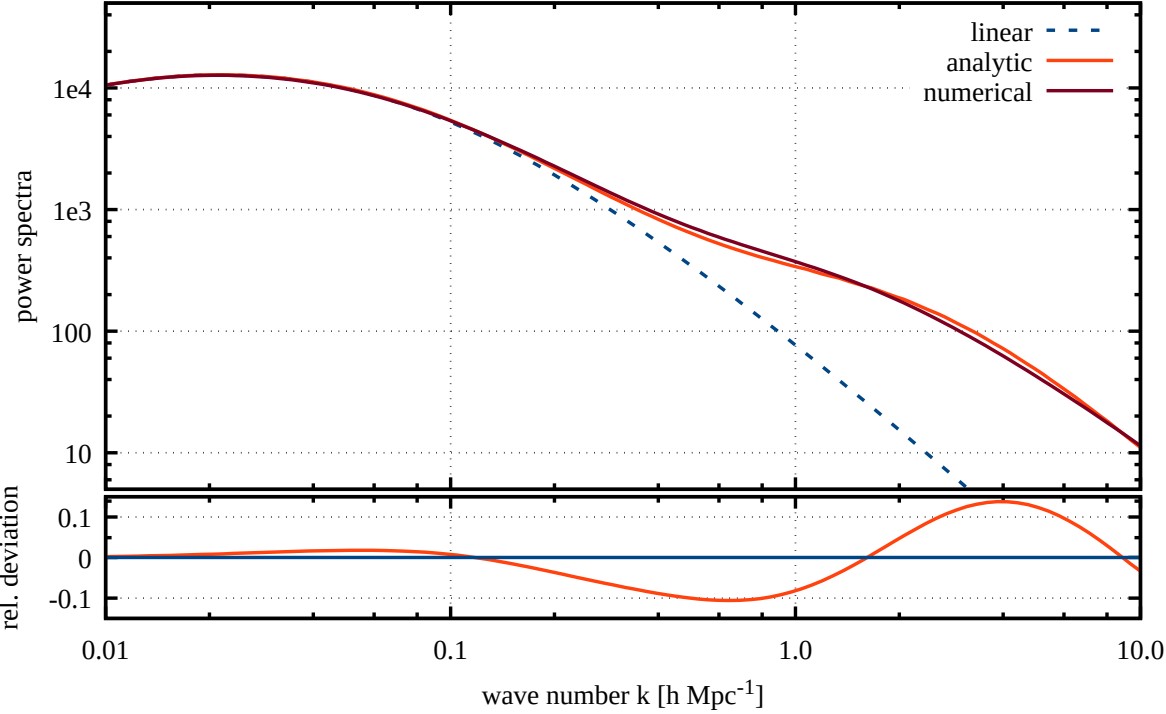

Figure 2: Analytic and numerical power spectra at redshift $z = 0$. The linear power spectrum is taken from [11], the numerical spectrum from [9]. The flat lower panel shows the relative deviation between the analytic and the numerical power spectra.

In this Figure, our analytic approximation (35) is compared at redshift $z = 0$ to the description by [9] of the power spectrum obtained from numerical simulations. As can be seen there, the analytic power spectrum agrees within typically $\lesssim 10\%$ with the numerical expectation up to wave numbers of $k \lesssim 10\,h\,\mathrm{Mpc}^{-1}$. With the parameters $\nu$ and $k_0$ self-calibrated from within KFT, the analytic expression (35) has no adjustable parameters since the Yukawa scale $k_0$ is set by KFT itself, and the viscosity is set to the square of the non-linear scale determined by the linear power spectrum from [11]. Expression (35) is also non-perturbative in the sense that the original exponential interaction operator is not expanded into a power series, but averaged in a mean-field approach.

We should emphasise that the derivation of the mean-field approximation is mathematically not fully rigorous, but in several steps guided by some intuition and the principles of statistical field theory. These are that we evaluate the mean interaction term with the linearly evolved power spectrum, reduce its damping by means of Burgers equation, and approximate the gravitational potential of the particles by a Yukawa form. Nonetheless, the agreement between our mean-field approximated analytic and the numerical results well into the non-linear regime of cosmic structure formation suggests that the microscopic approach of kinetic field theory, combined with a suitable choice for the inertial reference motion and adapting the effective force between particles to this reference motion, captures essential aspects of the physics of large-scale cosmic structure formation. The notorious shell-crossing problem does not occur in this approach, which is the main reason for the possibility to extend it far into the non-linear regime.

Instead of self-calibrating the two parameters $\nu$ and $k_0$ from within KFT, they can be considered as free parameters of the theory and chosen to optimise the agreement between numerical power spectra and the mean-field expression (35) by minimising the squared difference between them. Doing so, using the power spectrum from [9] as a reference, results in the power spectrum shown in Fig. 3.

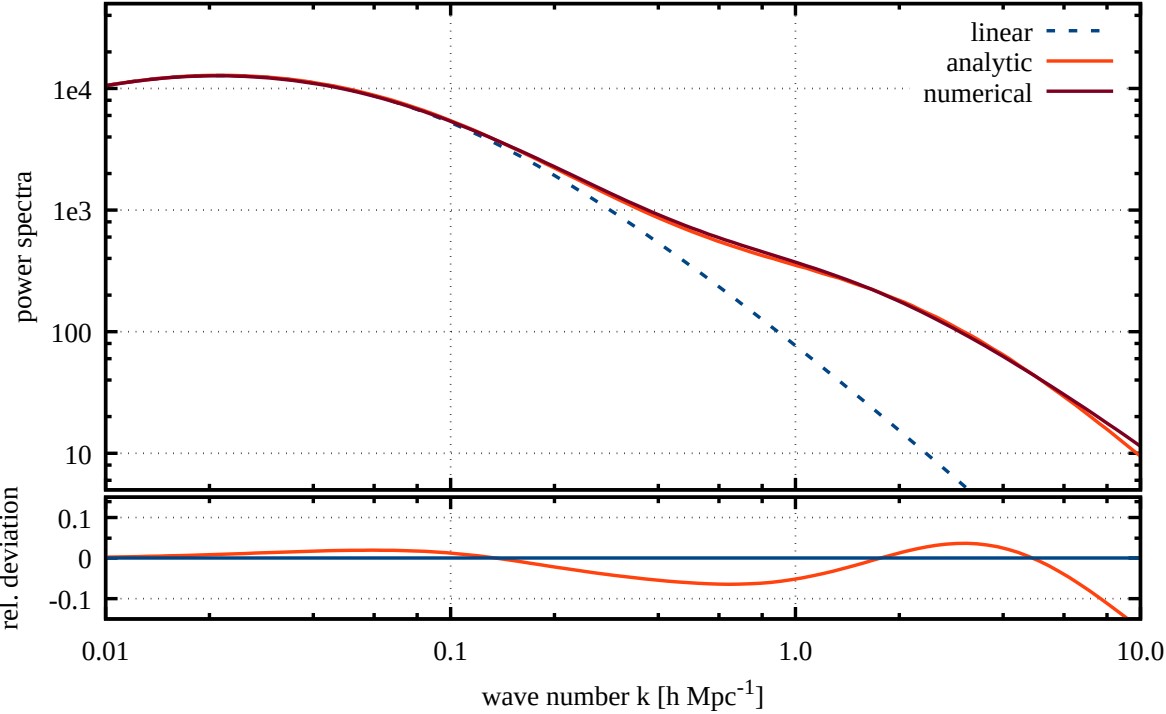

Figure 3: Like Fig. 2, but with the parameters $\nu$ and $k_0$ modified to optimise the agreement with the numerical results by [9].

The power spectrum shown there is obtained by reducing the Yukawa scale by 13% and increasing the displacement (48) by 6%. The relative deviation of the mean-field approximated, analytic power spectrum from its numerical counterpart is now lowered to typically $\lesssim 5\,\%$ up to $k \lesssim 10\,h\,\mathrm{Mpc}^{-1}$. For even smaller scales, the analytic power spectrum falls below the numerical expectation because then the mean-field approximation of the interaction term is no longer strong enough.

## 5   Summary and conclusion

The kinetic field theory for classical particle ensembles encapsulates the statistical information on the initial state and the propagator for the equations of motion in a generating functional which is closely analogous to the canonical or grand-canonical partition sum in thermodynamics. This generating functional evolves in time. Statistical macroscopic information is obtained from it by applying suitable operators. In this paper, we have used this approach to derive an analytic expression for the non-linear power spectrum of cosmic density fluctuations in a mean-field approximation of the particle-particle interaction term. Our main results are the closed, analytic approximation (35) for the non-linear power spectrum $P_\delta^{(\mathrm{nl})}$ and the expression (52) for the mean-field averaged interaction term. We have derived this form of the interaction term from KFT, averaging it over particle ensembles drawn from a statistically homogeneous, Gaussian random density field as shown in (38) and (42). This derivation is not mathematically rigorous because we have bypassed several complications and subtleties for the sake of simplicity. A rigorous assessment of the mean-field approximation needs to be based on systematic perturbation theory and will be worked out in a forthcoming paper. Nonetheless, the agreement with numerical results up to wave numbers $k \lesssim 10\,h\,\mathrm{Mpc}^{-1}$ is very good and encouraging.

It is important for our result that the microscopic, Hamiltonian equations of motion allow the introduction of an inertial motion with respect to the time $t = D_+ - 1$, corresponding to the celebrated Zel'dovich approximation, which captures the linear evolution of cosmic structures on large scales. Linearly growing, large-scale density fluctuations must then not exert any gravitational force on the inertial particle trajectories with respect to this time coordinate. This requires us to replace the Newtonian gravitational potential by an approximately Yukawa-shaped gravitational potential which ensures that only small-scale, non-linearly growing modes contribute to the particle-particle interaction. The Yukawa scale $k_0$ can be determined from kinetic field theory itself in a quickly converging iteration. We emphasise that the Yukawa shape is suggestive, but approximate and has no fundamental justification yet.

Our aim expressed in (35) to capture the non-linear evolution of the density-fluctuation power spectrum simply by a multiplicative, exponential interaction term applied to the linearly evolved power spectrum requires us to damp part of the interaction term on small scales. We do so by means of the damping term naturally appearing in the mean-field expression for the interaction term via KFT, but lowering the damping scale in a way derived from Burgers' equation. This introduces a viscosity parameter $\nu$, which is the square of a length scale characterising non-linear structures. A natural choice for this length scale is the non-linear radius defined in (44).

We thus have two parameters, $k_0$ and $\nu$, which can either be set by KFT itself or seen as free parameters. Self-calibrating both parameters with KFT leads to the mean-field approximated, non-linear power spectrum shown in Fig. 2, which already agrees well with numerical results. This agreement can further be improved by slightly adjusting both parameters, as shown in Fig. 3.

The initial state of the microscopic degrees of freedom is fully determined by the linear density-fluctuation power spectrum at the initial time, which can (and should) be set as early as the onset of

the matter-dominated epoch. We have used the cold-dark matter power spectrum here. Since we use the growth factor $D_+$ of linear density fluctuations as a time coordinate, the cosmological framework model enters only through the relation between redshift or scale factor and time, and through the time dependence of the effective particle mass. It can thus easily be generalised towards alternative dark-matter models, a different cosmological background, or alternative gravity theories. Non-linear cosmic power spectra such as these shown in Figs. 2 and 3 can be calculated within seconds on conventional laptops.

The approach followed in this paper can be improved in several ways. So far, we substantially simplified our mean-field approximation scheme, and we have modelled the particle-particle interaction potential by a Yukawa form for intuitive simplicity. The detailed form of the interaction potential could, however, also be derived from kinetic field theory itself; this would just cause the calculation of the mean interaction term to become more involved. The results shown should be seen as a further step towards a systematic interpretation of non-linear cosmic structures in terms of fundamental physics.

# Acknowledgements

We gratefully acknowledge fruitful discussions with many colleagues, most notably the always very helpful discussions with Manfred Salmhofer.

**Funding information**     This work was supported in part by Deutsche Forschungsgemeinschaft (DFG) under Germany's Excellence Strategy EXC-2181/1 - 390900948 (the Heidelberg STRUCTURES Excellence Cluster), by the Heidelberg Graduate School of Physics (HGSFP), by the Centre for Quantum Dynamics at Heidelberg University, and by a Technion Fellowship.

# A    Particle Trajectories

## A.1    Particle mass

Beginning with the effective particle mass $m$ defined in (4), the time derivative of $m$ is

$$\dot{m} = \frac{\mathrm{d}m}{\mathrm{d}D_+} = \frac{m'}{D'_+} \tag{54}$$

by definition of the time $t = D_+ - 1$. The prime denotes differentiation with respect to the scale factor $a$. We can insert (4) once more into (54) to arrive at

$$\dot{m} = \frac{a^3 E}{D'_+}\left[ D''_+ + \left(\frac{3}{a} + \frac{E'}{E}\right)D'_+ \right] . \tag{55}$$

The growth factor $D_+$ solves the linear growth equation. When transformed to the scale factor $a$ as an independent variable, this reads

$$D''_+ + \left(\frac{3}{a} + \frac{E'}{E}\right)D'_+ = \frac{3}{2}\frac{\Omega_{\mathrm{m}}}{a^2}D_+ . \tag{56}$$

Expressing the matter-density parameter $\Omega_{\mathrm{m}}$ by its value $\Omega_{\mathrm{m}}^{(i)}$ at the initial time,

$$\Omega_{\mathrm{m}} = \frac{\Omega_{\mathrm{m}}^{(i)}}{a^3 E^2} . \tag{57}$$

If we specify the initial conditions early in the matter-dominated epoch, we may further approximate $\Omega_{\mathrm{m}}^{(\mathrm{i})} \approx 1$. We thus find the expression

$$\dot{m} = \frac{3}{2} \Omega_{\mathrm{m}}^{(\mathrm{i})} \frac{a D_+}{m} \tag{58}$$

for the time derivative of the effective particle mass $m$. Inserting finally the potential amplitude $A_\varphi$ from the Poisson equation (5), we arrive at the time derivative

$$\dot{m} = m A_\varphi D_+ = m A_\varphi (t + 1) \tag{59}$$

of the effective particle mass $m$.

## A.2 Solution of the Hamiltonian equations of motion

The Hamiltonian equations of motion for the phase-space point $\vec{x} = (\vec{q}, \vec{p})^\top$ can be written in the form

$$\dot{\vec{x}} = A(t)\vec{x} - \begin{pmatrix} 0 \\ m\vec{\nabla}\varphi \end{pmatrix} \quad \text{with} \quad A(t) := \begin{pmatrix} 0_3 & m^{-1}\mathbb{1}_3 \\ 0_3 & 0_3 \end{pmatrix} . \tag{60}$$

Notice that $A(t)$ is a $6 \times 6$ matrix, with $0_3$ and $\mathbb{1}_3$ representing the zero and unit matrices in three dimensions, respectively. The homogeneous equation $\dot{\vec{x}} = A(t)\vec{x}$ is solved by $\vec{x}_{\mathrm{h}} = \exp[\bar{A}(t, 0)] \vec{x}_0$, with

$$\bar{A}(t, t') := \int_{t'}^t \mathrm{d}\bar{t}\, A(\bar{t}) = \begin{pmatrix} 0_3 & g_{\mathrm{H}}(t, t')\, \mathbb{1}_3 \\ 0_3 & 0_3 \end{pmatrix} , \quad g_{\mathrm{H}}(t, t') := \int_{t'}^t \frac{\mathrm{d}\bar{t}}{m(\bar{t})} . \tag{61}$$

Since $\bar{A}$ is nilpotent, $\bar{A}^2(t, t') = 0_6$, the homogeneous solution shrinks to

$$\vec{x}_{\mathrm{h}}(t) = \left(1 + \bar{A}(t, 0)\right) \vec{x}_0 . \tag{62}$$

By variation of the constant vector $\vec{x}_0$, the inhomogeneous equation of motion (60) leads to

$$\vec{x}_0(t) = \vec{x}^{(\mathrm{i})} - \int_0^t \mathrm{d}t' \left(1 + \bar{A}(t', 0)\right) \begin{pmatrix} 0 \\ m\vec{\nabla}\varphi \end{pmatrix} \tag{63}$$

and thus to the solution

$$\vec{x}(t) = \left(1 + \bar{A}(t, 0)\right) \vec{x}^{(\mathrm{i})} - \int_0^t \mathrm{d}t' \left(1 + \bar{A}(t, t')\right) \begin{pmatrix} 0 \\ m\vec{\nabla}\varphi \end{pmatrix} \tag{64}$$

for phase-space trajectories beginning at $\vec{x}^{(\mathrm{i})} = (\vec{q}^{(\mathrm{i})}, \vec{p}^{(\mathrm{i})})^\top$ at $t = 0$. The spatial trajectories are accordingly

$$\vec{q}(t) = \underbrace{\vec{q}^{(\mathrm{i})} + g_{\mathrm{H}}(t, 0)\vec{p}^{(\mathrm{i})}}_{=: \vec{q}_0(t)} - \int_0^t \mathrm{d}t'\, g_{\mathrm{H}}(t, t') m\vec{\nabla}\varphi =: \vec{q}_0(t) + \vec{y}_q(t) . \tag{65}$$

## A.3 Reference Trajectories

It is often convenient in cosmology to replace the force-free trajectories $\vec{q}_0(t) = \vec{q}^{(\mathrm{i})} + g_{\mathrm{H}}(t, 0)\vec{p}^{(\mathrm{i})}$ by the trajectories postulated in the Zel'dovich approximation,

$$\vec{q}_0(t) \to \vec{q}^{(\mathrm{i})} + t\vec{p}^{(\mathrm{i})} , \tag{66}$$

exchanging the Hamiltonian propagator $g_H(t, t')$ for $(t - t')$. The deviation $\vec{y}_q(t)$ defined in (65) as the difference between the actual and these reference trajectories is then determined by

$$\vec{y}_q(t) = -\int_0^t dt' \left[ g_H(t, t') m \vec{\nabla} \varphi + (1 - \dot{g}_H(t', 0)) \vec{p}^{(i)} \right] . \tag{67}$$

Implicitly defining an amplitude $A_p(t')$ by

$$\int_0^t dt' \, (1 - \dot{g}_H(t', 0)) \overset{!}{=} \int_0^t dt' \, g_H(t, t') A_p(t') , \tag{68}$$

we can write (67) in the form

$$\vec{y}_q(t) = -\int_0^t dt' \, g_H(t, t') \left( m \vec{\nabla} \varphi + A_p(t') \vec{p}^{(i)} \right) . \tag{69}$$

Since the initial momentum is the gradient of a potential, $\vec{p}^{(i)} = \vec{\nabla} \psi$, (69) suggests defining an effective potential

$$\phi := \varphi + \frac{A_p}{m} \psi \tag{70}$$

such that

$$\vec{y}_q(t) = -\int_0^t dt' \, g_H(t, t') m \vec{\nabla} \phi . \tag{71}$$

Differentiating (68) twice with respect to the time $t$ and using $\dot{g}_H(t, t') = m^{-1}(t)$ gives

$$A_p(t) = \dot{m} = m A_\varphi D_+ \tag{72}$$

with (59), and thus the trajectories

$$\vec{q}(t) = \vec{q}^{(i)} + t\vec{p}^{(i)} - \int_0^t dt' \, g_H(t, t') \, m \vec{\nabla} \phi , \quad \phi = \varphi + A_\varphi D_+ \psi . \tag{73}$$

Continutity demands that the initial velocity potential satisfies the Poisson equation $\vec{\nabla}^2 \psi = -\delta^{(i)}$.

## A.4 Unifying Propagators

It is often convenient to replace the propagator $g_H(t, t')$ in (73) also by the time difference $t - t'$. For doing so, we implicitly introduce an effective force $\vec{f}$ demanding

$$\int_0^t dt' \, g_H(t, t') m \vec{\nabla} \phi \overset{!}{=} -\int_0^t dt' \, (t - t') \, \vec{f}(t') , \tag{74}$$

and solve for $\vec{f}(t)$. Differentiating (74) twice with respect to $t$, we find

$$\vec{f}(t) = -\vec{\nabla} \phi + \frac{\dot{m}}{m^2} \int_0^t dt' \, m \vec{\nabla} \phi . \tag{75}$$

With this expression for $\vec{f}(t)$, we can write the spatial trajectories as

$$\vec{q}(t) = \vec{q}^{(i)} + t\vec{p}^{(i)} + \int_0^t dt' \, (t - t') \vec{f}(t') . \tag{76}$$

Finally replacing $\vec{x} = (\vec{q}, \vec{p})^\top$ by $(\vec{q}, \dot{\vec{q}})^\top$, we can bring the solution $\vec{x}(t)$ of the equations of motion into the form

$$\vec{x}(t) = G(t, 0)\vec{x}^{(\mathrm{i})} + \int_0^t \mathrm{d}t' \, G(t, t') \begin{pmatrix} 0 \\ \vec{f}(t') \end{pmatrix} \tag{77}$$

with the $6 \times 6$ matrix

$$G(t, t') = \begin{pmatrix} \mathbb{1}_3 & (t - t') \, \mathbb{1}_3 \\ 0_3 & \mathbb{1}_3 \end{pmatrix} . \tag{78}$$

# B  Convolving the particle-particle force with the power spectrum

Based on the result (75) for the effective force, we begin with the expression

$$\vec{f}_{\mathrm{p}} = -\vec{\nabla}v + \frac{\dot{m}}{m^2} \int_0^t \mathrm{d}t' \, m\vec{\nabla}v \tag{79}$$

for the particle-particle force represented by the potential $v$. The averaged interaction term (51) requires projecting the Fourier-transformed particle-particle force convolved with the power spectrum onto the wave vector $\vec{k}$. With this in mind, we first convolve the Fourier-transformed potential gradient $\widetilde{\nabla v} = \mathrm{i}\vec{k}\tilde{v}$ with the damped power spectrum. Inserting $\tilde{v}$ from (14) then leads to the intermediate equation

$$\vec{k} \cdot \left( \widetilde{\nabla v} * \bar{P}_\delta \right)(k) = -\frac{\mathrm{i}A_\varphi}{n} \int_{k'} \frac{\vec{k} \cdot \left( \vec{k} - \vec{k}' \right)}{k_0^2 + \left( \vec{k} - \vec{k}' \right)^2} \bar{P}_\delta (k') . \tag{80}$$

We introduce spherical polar coordinates to evaluate the integral in (80) and turn the coordinate system such that $\vec{k}$ points into the direction of the polar axis. Defining the cosine $\mu$ of the polar angle between $\vec{k}$ and $\vec{k}'$, further substituting $\kappa := k'/k$ and $\kappa_0 := k_0/k$ then leads to

$$\vec{k} \cdot \left( \widetilde{\nabla v} * \bar{P}_\delta \right)(k) = -\frac{\mathrm{i}A_\varphi}{n} \frac{k^3}{(2\pi)^2} \int_0^\infty \mathrm{d}\kappa \, \kappa^2 \, \bar{P}_\delta \left( k\kappa, t' \right) \, J(\kappa, \kappa_0) \tag{81}$$

with

$$J(\kappa, \kappa_0) := \int_{-1}^1 \mathrm{d}\mu \, \frac{1 - \kappa\mu}{1 + \kappa_0^2 + \kappa^2 - 2k\kappa\mu} = 1 + \frac{1 - \kappa^2 - \kappa_0^2}{4\kappa} \ln \frac{\kappa_0^2 + (1 + \kappa)^2}{\kappa_0^2 + (1 - \kappa)^2} . \tag{82}$$

The left panel of Fig. 4 shows the function $J(\kappa, \kappa_0)$ for different values of $\kappa_0$. It falls off $\propto \kappa^{-2}$ for $\kappa \gg 1$ and tends towards the constant $2/(1 + \kappa_0^2)$ for $\kappa \ll 1$. The function $J(\kappa, \kappa_0)$ can thus be seen as a filter function for the power spectrum. We introduce the moment

$$\sigma_J^2 := \frac{k^3}{(2\pi)^2} \int_0^\infty \mathrm{d}\kappa \, \kappa^2 \, \bar{P}_\delta^{(\mathrm{i})}(k\kappa) \, J(\kappa, \kappa_0) \tag{83}$$

of the damped *initial* power spectrum $\bar{P}_\delta^{(\mathrm{i})}$, filtered with the function $J$, and bring (81) into the form

$$\vec{k} \cdot \left( \widetilde{\nabla v} * \bar{P} \right)\!\left( \vec{k} \right) = -\frac{\mathrm{i}A_\varphi}{n} D_+^2 \sigma_J^2 = -\frac{\mathrm{i}}{n} \frac{\dot{m}}{m} D_+ \sigma_J^2 . \tag{84}$$

In the last step, we have used the time derivative $\dot{m}$ from (59) in Appendix A. According to (79), and using (84), the convolved force $\tilde{f}_{\mathrm{p}} * \bar{P}$ projected on the wave vector $\vec{k}$ is

$$\vec{k} \cdot \left( \tilde{f}_{\mathrm{p}} * \bar{P} \right) = \frac{\mathrm{i}}{n} \frac{\dot{m}}{m} \left[ D_+ \sigma_J^2 - \frac{1}{m} \int_0^t \mathrm{d}t' \, \dot{m} D_+ \sigma_J^2 \right] . \tag{85}$$

With this result, we return to the averaged interaction term (51), finding the averaged interaction term (52).

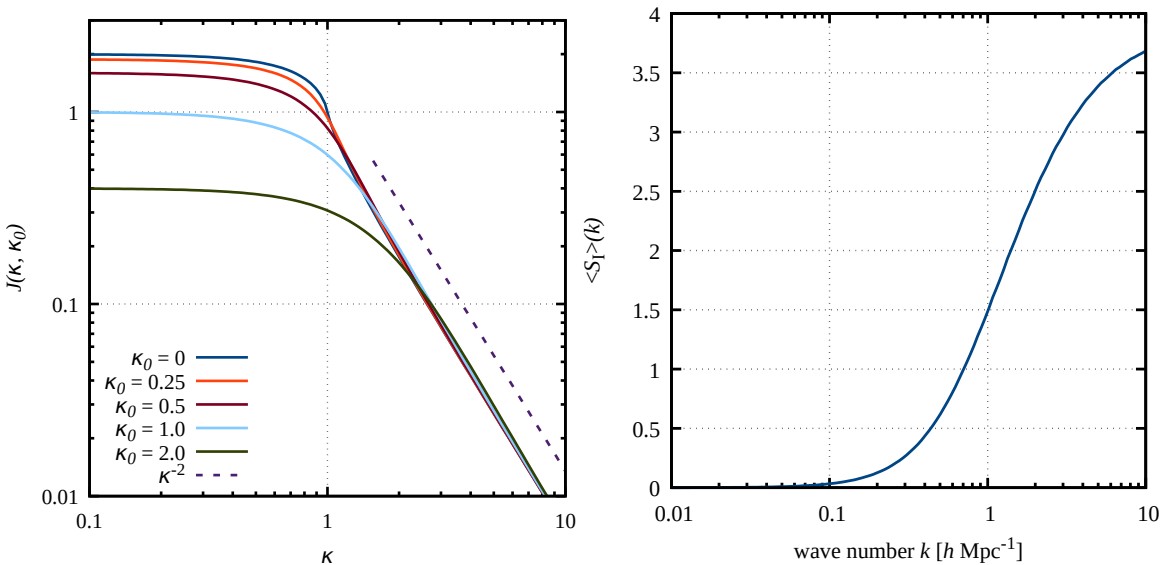

Figure 4: *Left panel*: The function $J(\kappa, \kappa_0)$ appearing in the averaged force term $\langle \tilde{f}_{12} \rangle$ is shown here for five values of $\kappa_0$. While the asymptotic behaviour of $J(\kappa, \kappa_0) \propto \kappa^{-2}$ for $\kappa \gg 1$ is unaffected by $\kappa_0$, $J(\kappa, \kappa_0)$ approaches the value $2/(1 + \kappa_0^2)$ for $\kappa \ll 1$. *Right panel*: Average interaction term $\langle S_{\mathrm{I}} \rangle (k)$ as a function of the wave number $k$.

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
