# Peer review of "Kinetic field theory: Non-linear cosmic power spectra in the mean-field approximation"

_SciPost Physics_

## Round 1 · Referee Report · Anonymous (Referee 1) · 2021-2-21

Strengths

A new schema to model nonlinear power spectrum

Weaknesses

Not rigorous, no properly setout frame work

Report

This works report on a new way of modeling nonlinear power spectrum in cosmology. The authors themselves acknowledge that their method is not rigorous.

I can recommend the paper for publication, after improvements/changes suggested below. My main reason for acceptance is that the paper provides a new different approach, to model nonlinear power spectrum in cosmology and as such is interesting and enrichens the field. The paper is well-written and although I have not checked author's calculations line-by-line, I trust there is no major problem with the maths. My concern is with the physics at the foundation of their work and these are the problems that I wish to ask the authors to address.

Requested changes

(1)
The authors refer to their method as KFT: however I do not see anything truely "kinetic" , in the statistical physics sense of the word, in their work. No mention of distribution function neither of the Boltzmann equation. In fact they use fluid equations such as Burger's model to describe various phenomena. I would ask the authors to either relate to the established kinetic theory in statphys or remove the term.
(2)
They claim that the interaction between particles, past Zeldovich crossing, is Yukawa rather than Newton. I ask the authors to tone this down: the claim is that the Newtonian interaction can be "modeled" by Yukawa potential after orbit crossing. They are not suggesting that we run cosmological simulations and rewrite our perturbation theory using Yukawa potential !! I ask the authors to rewrite the text concerning this point.
(3) Yukawa coupling: when used in particle physics has a proper well-defined parameter in terms of fundamental constants and masses of particles. There is no such a things here: the Yukawa coupling scale is rather adhoc, it is time-dependent (which the authors just ignore) and itself a function of density fluctuation, etc. This is related to the general problem in cosmology that we lack a proper "perturbation parameter". I ask the authors to address these issues and to justify the fact that they are ignoring the time-dependence of their "Yukawa scale".

(3) Burger Turbulence. The authors refer to Burger Turbulence as Navier-Stokes, I ask the authors to correct this in the text, as Burger turbulence is just a model of potential turbulence and has little to do with Navier-Stokes equation. I find the use of Burger turbulence magical rather than scientific ! To search for a daming scale they use Burgers, also modified slightly because in Burgers there are shocks and so no smoothed-out structures. This is more a demonstration of the failure of their Yukawa modeling of the particle interaction than anything else. Indeed if Yukawa potential could model the particle interactions properly, there would be no need to then invoke yet another model (fluid and not kinetic) to model again the particle interactions in the multistream regions.
Indeed Burger turbulence has been used extensively years ago to model large-scale structure of the Universe and was refered to as "adhesion model". Since the purpose of this work is to model nonlinear power spectrum, then I ask the authors to also compare their power spectrum to those from the adhesion model, and indeed other existing models of nonlinear power spectrum. I ask the authors to provide plot on which they show the PS from simulations,from their model and also adhesion model, truncated ZA model, models of Scoccimarro et al and Taruya et al, Ostriker et al etc...
(4)
The authors test their model against simulations and show that indeed their model provides a good agreement for the nonlinear PS. However they need use simulations to set their Yukawa scale and viscosity parameters in the first place ! Indeed their free parameters are best determined by requiring their model to match the simulations. This problem need be addressed. Apart from this, I find it rather disappointing that the numerical simulations provide a final test of the "kinetic" model here. Indeed the real interest of using kinetic theory is that they can go beyond numerical simulations which are limited by resolution. Kinetic theory could model structures that are often missed in numerical simulations because of numerical artifacts. In this resepect kinetic theory remains superior to present numerical simulations and only a full simulation of Vlasov-Poisson (VP) equation would provide the ultimate model of nonlinear evolution. Since VP simulation are non-existent, and shall most probably not emerge in the forseeable future because of the huge computational resources they require, a clever kinetic theory-based model could provide a guideline in the meantime. However disappointingly this point is completely missed out by the authors. Authors need discuss this problem, namely can they model structures 'caustics, streams, that are often missed in the simulations due to lack of resolution ?

---

## Round 1 · Referee Report · Anonymous (Referee 2) · 2021-3-12

Strengths

  1. This paper presents in a concise and clear way the results of an approach which should be seen as complementary to numerical simulations and alternative to other techniques currently used, such as the Effective Field Theory of LSS (EFT-of-LSS). I think it is absolutely crucial to have alternative approaches like this on the market.
  2. The results obtained here are simple to understand and derive from a clear set of assumptions and approximations.
  3. The connection to Burgers turbulence is well motivated and indicates an important connection with previous work on understanding the dynamics of LSS. This is an important merit of this approach which is completely absent in e.g. the EFT-of-LSS approach.

Weaknesses

I don't see specific weaknesses of the current paper.

Report

I believe that the paper presents good evidence for the robustness of the approach and provides simple and clean derivation of the results. I thus recommend the paper for publication in SciPost.

Requested changes

No mandatory changes are required. I would only suggest a few more sentences in the introduction to recap the fundamental basis underlying the KFT approach and the motivation for the used of the word "Kinetic" in the method.

---

## Editorial Decision

resubmitted